# Survey on Sternal Wound Management in the Italian Pediatric Cardiac Intensive Care Units

**DOI:** 10.3390/healthcare9070869

**Published:** 2021-07-09

**Authors:** Angela Prendin, Benedicta Tabacco, Paola Claudia Fazio, Veronica Strini, Luca Brugnaro, Ilaria De Barbieri

**Affiliations:** 1Independent Researcher, 35100 Padua, Italy; 2Independent Researcher, Kirkland, WA 98034, USA; benedicta.tabacco.89@gmail.com; 3Pediatric Intensive Care Unit, University-Hospital of Padua, 35100 Padua, Italy; paolaclaudia.fazio@aopd.veneto.it; 4Clinical Research Unit, University-Hospital of Padua, 35100 Padua, Italy; veronica.strini@aopd.veneto.it; 5Training and Development Unit of the Health Professions, University-Hospital of Padua, 35100 Padua, Italy; luca.brugnaro@aopd.veneto.it; 6Nurse Coordinator Woman’s & Child’s Health Department, University-Hospital of Padua, 35100 Padua, Italy; ilaria.debarbieri@aopd.veneto.it

**Keywords:** surgical wound infection, cardiac surgical procedures

## Abstract

(1) Background: a review of the literature found a lack of standardized pediatric guidelines regarding wound management after cardiac surgery. (2) Objective: the aim of the study is to investigate the cardiac surgical wound management in Italian pediatric cardiac intensive care units. (3) Methods: we sent an online questionnaire to the 13 Italian pediatric cardiac intensive care units. (4) Results: ten pediatric cardiac intensive care units (77%) have a protocol for the management of the cardiac surgical wound. The staff members that mainly have the responsibility for the wound management after cardiac surgery are registered nurses and physicians together both in the pediatric cardiac intensive care units (69%), and when a patient is transferred to another ward (62%). Thirty-eight percent of the pediatric cardiac intensive care units have a protocol used to monitor wound infection, and the staff mostly uses a written shift report (54%) to monitor the infection. (5) Discussion: this is the first survey to investigate the management of the wound after cardiac surgery in Italian pediatric cardiac intensive care units. The small sample size and the fact that the centers involved are only Italian cardiac intensive care units are the limits of this study. (6) Conclusions: in the Italian pediatric cardiac intensive care units it emerged that there is a diversity in the treatments adopted and a lack of specific protocols in the management of the pediatric cardiac surgical wound.

## 1. Background

Surgical Site Infections (SSIs) are a possible serious complication after cardiac surgery and are associated with increased morbidity (antibiotic use, a second surgery, prolonged hospitalization, prolonged periods of mechanical ventilation, inotropic support) and mortality [1,2]. The incidence of SSIs after pediatric cardiothoracic surgery in the USA is reported to be between 0.25% and 6%, and it has an associated mortality from 7% to 20% [3]. Pediatric recommendations are founded on expert opinions and from data extrapolated from adult guidelines [4]. Adult guidelines are based on risk factors such as obesity, diabetes, and smoking, which are not the same risk factors considered valid for developing SSIs in children [5]. Potential pre-operative, intra-operative, and post-operative risk factors for children include: age lower than one month, the duration of surgery, presence of genetic abnormalities, prolonged extracorporeal circulation time (ECMO), delayed sternal closure, pre-operative hospitalization and prolonged post-operative hospitalization, post-operative hemorrhage, and persistent low cardiac output [1,6,7]. From the literature it has emerged that a consistent number of studies sustained the bundle approach as a valid instrument for reducing the instances of SSIs [2,6,8,9]. Another four studies supported pediatric preventive guidelines and practice bundles as effectual for preventing SSIs [5,10,11,12]. Woodward et al. reported the experience of their center using an evidence-based protocol. The implementation and use of this protocol reduced the SSI rate by 64% between the first and second year of the analyzed study [8]. Even if the majority of those are single-center studies, the benefits of these projects and instruments are real and concrete. We wanted to know the reality in our country, and we wanted to investigate in the Italian Cardiac Intensive Care Units (CICUs) that accept pediatric patients the management and the sensibility about this possible severe complication that is the surgical site infection after cardiac surgery in the pediatric population. 

## 2. Objective of the Study

The aim of this study was to evaluate the management of the wound dressing after cardiac surgery in critically ill children admitted into Italian CICUs.

## 3. Methods

Between May 2018 and May 2019, we completed a survey contacting the nurse coordinators of the 13 Italian CICUs, through a telematic questionnaire (Appendix A). We wanted to investigate the management of the wound after cardiac surgery in critically ill children admitted in these units. Our questionnaire contained 10 multiple choice questions inquiring about:the characteristics of the CICUs (accepting neonatal and pediatric or neonatal, pediatric, and adult patients; the number of beds for pediatric patients and pediatric cases during the year 2017);the presence of a standardized protocol for the management of the cardiac wound after cardiac surgery;in the case of a standardized protocol, which staff member has the responsibility of monitoring and dressing management;the presence of a standardized protocol used to monitor the possible infection of the wound after cardiac surgery;in the case of a standardized protocol, how the staff takes note of the presence of the infection and its treatment;in the case of the patient’s transfer to another ward, which staff member has the responsibility of monitoring and dressing management.

There was no need for the involvement of the ethics committee since the work involved a descriptive survey. A questionnaire was distributed to 13 nurse coordinators of Italian CICUs. Two PICU registered nurses developed the questionnaire based on their experience. The aim of these 10 questions was to investigate the management and responsibility for monitoring and dressing pediatric cardiac surgery wounds in Italian CICUs. The results were anonymously saved, and it was set a numerical identifier to maintain the confidentiality of each unit. The data has been processed through a descriptive analysis. 

## 4. Results

This study includes 13 CICUs. All the centers answered the questionnaire. A total of 69% of the CICUs accept neonatal, pediatric, and adult patients, while 31% of them accept neonatal and pediatric clients (Table 1). Ten CICUs have a fixed number of beds for pediatric patients while three units do not have a set number of beds for children undergoing cardiac surgery. The average number of pediatric cases in 2017 for all of the 13 cardiac ICUs was 269.9. Two centers out of thirteen did not respond to this specific question, and one CICU shared the number of cases for the adult and pediatric population together. Ten units (77%) claimed to have adopted a protocol for the management of the wound after cardiac surgery. Eight out of ten units (a total of 62%) enclosed their protocol with their answers to the questionnaire (Figure 1a). Analyzing them we found that only four units are actual protocols, of which just one is pertinent to the covered topic; two are operative instructions; and two are operative surveillance. Nevertheless, none of them are specific for the management of the cardiac wound in the pediatric population: seven argue about surgical wounds in general and referring to the adult population, and one elucidate the external ventricular leads and mediastinitis care. All the centers that do not have a protocol (23%) agree that they would like to have one. When a standardized protocol is used in the CICU, the monitoring and dressing management is either a responsibility of: the registered nurse (RN) (31%), the cardiothoracic surgeon (0%), the CICU physician (0%), or the RN and physician (cardiothoracic surgeon, CICU physician, or multidisciplinary team) together (69%) (Figure 2a). Of the CICUs that responded, 38% have a protocol for monitoring if a wound infection is present after cardiac surgery, while 62% of the CICUs declared that they do not have one (Figure 1b). The monitoring of the wound infection after cardiac surgery occurs through: a written shift report on the patient’s medical record (54%), a validated rating scale (of which one center utilizes a regional infection surveillance) (38%), or a verbal shift report (0%). In the category “Other” (8%) evaluate the presence of the infection through discussion amongst a multidisciplinary team. When the patient is transferred to another ward, the wound is either managed by: the RN (15%), the cardiothoracic surgeon (15%), or the RN and the unit physician together (62%). In the “Other” category (8%) the interviewees specified that the wound is managed by “RN and collegiate evaluation” (Figure 2b).

## 5. Discussion

To the best of our knowledge, this is the first survey to investigate the management of the wound after cardiac surgery in the Italian CICUs accepting pediatric patients. The organizational structure is different in each CICU regarding the type of patients they accept, neonatal and pediatric, or neonatal, pediatric and adult, and the number of beds allocated for pediatric cases undergoing cardiac surgery. Eight units shared the documentation that they follow for the management of the cardiac wound. Although the documents provided by the eight CICUs are not on-target for the cardiac pediatric population, in four of them, there are indications about the timeline for the disinfection of the incision: one unit cleans the wound 24 to 48 h after the surgery, two CICUs remove the dressing of the first intention wounds 24 to 48 h after the surgery, and one unit removes the dressing of the first intention wounds after 48 h. Scientific evidence recommends that the sternal dressing should stay in place for the first 48 h after sternal closure, and it needs to be replaced only if visibly dirty [6,7,13]. After 48 h, the dressing should be removed [8]. Specific solutions for the disinfection of the surgical wound are listed in these documents provided by the interviewed CICUs: the povidone–iodine solution (the most commonly used), chlorhexidine, hydrogen peroxide, and Amuchina. Although in the literature there are no precise indications about the most appropriate solution for the disinfection of the surgical wound [14], detailed prophylaxis exists that specify pre- and intra-operative recommendations [2,7,9,12,15]. Regarding the antibiotic prophylaxis, Bath et al. (2016) demonstrated that limiting antimicrobial prophylaxis to 48 h after pediatric cardiac surgery neither increase the incidence of SSIs nor alter other clinical outcomes in this population. Furthermore, cefazolin and cefuroxime are equally recommended as first-line agents for antimicrobial prophylaxis in cardiac surgery. Murray et al., (2014) identified that limiting the peri-operative antibiotic prophylaxis after cardiac surgery to 48 h in neonates with a closed sternum and 24 h after sternal closure is safe and does not increase the rate of SSI. Another fundamental aspect that deserves further studies is to establish the best timing for administering pre-operative antibiotics and antiseptic skin preparation [7,8]. Some studies suggest the use of the pre-operative nasal culture [6,15,16,17,18]. However, further studies are needed in order to analyze this association. Less than half of the centers reported having a protocol for monitoring the wound infections after cardiac surgery, and only one center uses regional infection surveillance. This regional guideline is based on the European surgical site infection surveillance protocol (HAI-SSI), which was defined by the European Center for Diseases Prevention and Control (ECDC). This protocol classifies the procedures in diverse categories of intervention according to the US National Healthcare Safety Network (NHHSN). The clinical similarity of the surgical procedures belonging to these categories leads to homogeneous comparisons [19,20,21]. From the literature [5,10,11,12], it is clear that the implementation of quality improvement measures, such as linkage of registry and infection control surveillance, pediatric preventive guidelines and practice bundles, is considered an effective and systematic way to deliver preventive measures of SSIs. Woodward et al. (2012) proposed a hospital wide quality improvement project utilizing a protocolized approach in child cardiac surgery. This protocol was developed using evidence-based- and best practices from the adult and pediatric literature, and it was associated with a decreased number of infections in children. Analyzing the answers collected from this Italian survey, it can be said that in the majority of cases the staff members that have the responsibility for the wound management after cardiac surgery are RNs and physicians together. The analysis of the literature confirms that multidisciplinary work represents a fundamental point in the wound management (Cannon et al., 2016).

### Limitations

A limit of this survey is the use of a questionnaire that was created specifically for this research; it was not previously validated. Furthermore, not all the CICUs sent their protocol on the pediatric cardiac surgical wound management, and the documentation sent was in the large part not directly connected to the aim of this study. Ultimately, we are aware of the small sample size and that the centers involved are only Italian CICUs.

## 6. Conclusions

Analyzing the data collected from this study, we can say that the Italian CICUs that accept pediatric patients follow different protocols for the management of the cardiac surgical wound, and a national standardized protocol or guideline to follow does yet not exist. Nevertheless, in the majority of cases registered nurses and physicians share the responsibility for wound dressing and care. There are diverse fundamental points that can have a positive impact on the cardiac surgical wound outcome, such as the best timing for the administration of antibiotic prophylaxis, the dressing management, and the collaboration between healthcare professionals. The final aim is the reduction in the incidence of SSIs. Further research is necessary for the realization of a national standardized protocol or guideline that incorporates all of these points.

## Figures and Tables

**Figure 1 healthcare-09-00869-f001:**
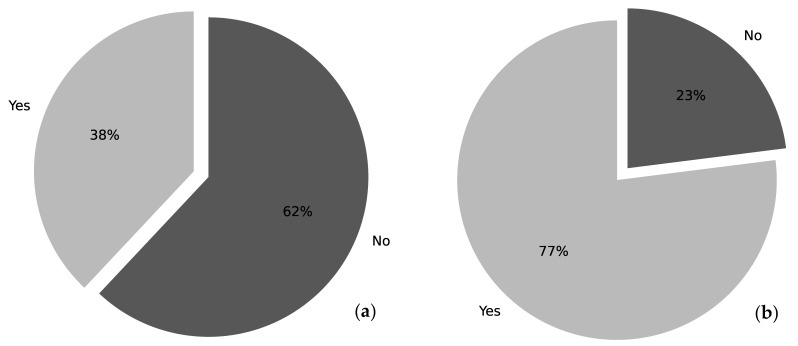
Standardized protocols used in CICUs. (**a**) Presence of a standardized protocol for the management of cardiac surgical wound. (**b**) Presence of a standardized protocol for monitoring the cardiac surgical wound infection.

**Figure 2 healthcare-09-00869-f002:**
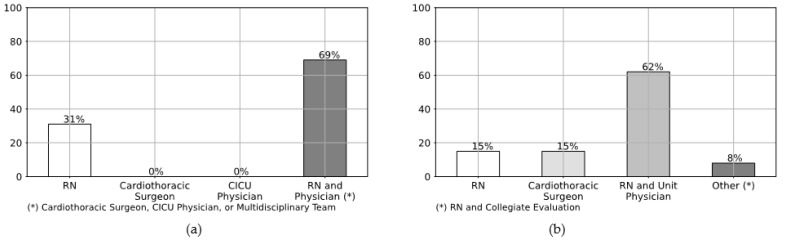
Monitoring and dressing management of the cardiac surgical wound. (**a**) Staff member in the CICU who is responsible for monitoring and dressing management. (**b**) Staff member who is responsible for monitoring and dressing management in case of patient’s transfer to another ward.

**Table 1 healthcare-09-00869-t001:** Characteristics of the studied pediatric cardiac intensive care units (CICUs).

CICUs IDs	CICUs Type	Number of Beds for Cardiac Pediatric Patients	Number of Cardiac Pediatric Cases in 2017
1	N/P	4	202
2	N/P	5	80
3	N/P + A	7	260
4	N/P + A	3	90
5	N/P	16	-
6	N/P + A	3	~200
7	N/P + A	16	594
8	N/P + A	4	215 *
9	N/P + A	4	270
10	N/P + A	4	33
11	N/P	19	-
12	N/P + A	9	550
13	N/P + A	7	350 **

N/P: neonatal/pediatric; A: adult; -: unspecified number of admissions. * P and A cases altogether; ** P and GUCH (grown up congenital heart) cases altogether.

## Data Availability

Data is contained within the article.

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
