# Peer review of "Survey on Sternal Wound Management in the Italian Pediatric Cardiac Intensive Care Units"

_healthcare, 2021, doi:10.3390/healthcare9070869_

Round 1

Reviewer 1 Report

The work is totally justified because there are few data in the pediatric literature and the guidelines are extrapolated from adults.

They describe a study about the management of the surgical wound, but the results only show the data on how to evaluate the wound by the medical team that cares for both wounds without or with infection. It would be nice if they included in their results the data on the use of local disinfectants and the type of dressing used (cloth, paper, silicone, etc.).

These data would give a broader and more complete view of surgical wound management. In fact the authors speak of these concepts in the discussion, but not in results. My advice is to include the data they show at the Discussion in the Results section and then comment on them in the Discussion.

Methods title section is misspelled

On page 3.9, line should say:… 62% of the CICUs declares not to have… instead of… 62% of the CICUs declares to not have…

In the graphs of figure 2 the first columns of each of them are missing.

In line 7 of the discussion it says: “these documents are a protocol” and it should say “these documents are the same protocol” or “these documents are a single protocol”.

Author Response

REVIEW 1

-They describe a study about the management of the surgical wound, but the results only show the data on how to evaluate the wound by the medical team that cares for both wounds without or with infection. It would be nice if they included in their results the data on the use of local disinfectants and the type of dressing used (cloth, paper, silicone, etc.).

Unfortunately the eight protocols that we have been calling for are not specific about the type of dressing or disfectant to use during dressing. This survey wants a first approach to understand how this procedure is carried out, at national level. The next step is to standardize this procedure, through a comparison with international CICUs.

-These data would give a broader and more complete view of surgical wound management. In fact the authors speak of these concepts in the discussion, but not in results. My advice is to include the data they show at the Discussion in the Results section and then comment on them in the Discussion.

In the results part we briefly described the eight protocols included. Unfortunately none of them refer to the population and pediatric and they are not even specific on wound management or on the prevention of infections starting from the intervention.

-Methods title section is misspelled.

Ok thanks, that was corrected.

-On page 3.9, line should say:… 62% of the CICUs declares not to have… instead of… 62% of the CICUs declares to not have…

Ok thanks, that was corrected.

-In the graphs of figure 2 the first columns of each of them are missing.

Thanks, we fixed it by inserting the missing columns.

-In line 7 of the discussion it says: “these documents are a protocol” and it should say “these documents are the same protocol” or “these documents are a single protocol”.

We have clarified the sentence better, that is, they are three distinct protocols but only one concerns the research topic.

REVIEW 2

-There are grammatical error which are minor.  Description of the wound care protocols would be helpful.  Future plans for evaluation or creating protocols for an example treatment regimen.  The survey results have some validity as they describe an inconsistent practice compared between multiple, dissimilar CVICUs.  It is not real ground breaking information but could be used for future work. 

We have made grammar corrections.

The purpose of this research was to have a national overview of this procedure and after carry out further research to try to standardize this practice.

Reviewer 2 Report

There are grammatical error which are minor.  Description of the wound care protocols would be helpful.  Future plans for evaluation or creating protocols for an example treatment regimen.  The survey results have some validity as they describe an inconsistent practice compared between multiple, dissimilar CVICUs.  It is not real ground breaking information but could be used for future work. 

few grammatical corrections needed, presentation of protocols from CICU would be helpful

Author Response

REVIEW 2

-There are grammatical error which are minor.  Description of the wound care protocols would be helpful.  Future plans for evaluation or creating protocols for an example treatment regimen.  The survey results have some validity as they describe an inconsistent practice compared between multiple, dissimilar CVICUs.  It is not real ground breaking information but could be used for future work. 

We have made required grammar corrections.

Exactly, the data collected have shown a discrepancy in the management of the wound in the pediatric population. We wanted to obtain a nation overview and present the data. A standardized protocol is a further step that we would like to pursue.